# Understanding the Experiences of People Living with Stroke Engaging in a Community-Based Physical-Activity Programme

**DOI:** 10.3390/healthcare11020154

**Published:** 2023-01-04

**Authors:** Matthew Smith, Andrew Scott, Serena Mellish, James Faulkner

**Affiliations:** 1Faculty of Health and Wellbeing, School of Sport, Health, and Community, University of Winchester, Winchester SO22 4NR, UK; 2School of Sport, Health and Exercise Science, University of Portsmouth, Portsmouth PO1 2ER, UK

**Keywords:** stroke, exercise, interviews, qualitative methods

## Abstract

Research has evidenced that regular exercise can provide physical and physiological benefits for people living with stroke. Our study aims to explore the experiences of people living with stroke when participating in a community physical activity programme. This programme was created to offer targeted physical activity and education interventions following the discharge of patients from the healthcare pathway. This qualitative study involved semi-structured interviews with 16 participants living with stroke who were recruited from individuals who had engaged with the activity programme. A reflexive thematic analysis was conducted on the data, and four overarching themes were developed: (i) Feelings of appreciation, (ii) Interactions with other patients, (iii) Positive contributions of trained instructors, and iv) Personal progress. Generally, participants reported very positive perceptions of the exercise programme, and were very grateful for the opportunity that the exercise classes provided. We hope that these findings will offer practical suggestions for healthcare providers who might develop similar activity programmes for clinical populations.

## 1. Introduction

For individuals living with stroke, research has shown that regular exercise can provide physiological benefits such as improvement in vascular health measures (e.g., blood pressure, blood lipid profile), aerobic fitness [1,2] and strength, balance and gait [3], as well as improvements in psychosocial health outcomes [4,5]. In general, exercise has been shown to improve engagement in activities of daily living and quality of life for individuals living with stroke (see [6] for a review), and maintaining a physically active lifestyle is therefore proposed to be very beneficial for this population. However, in the United Kingdom (UK)high levels of stroke survivors do not meet recommended physical activity guidelines [7]. Clinical guidelines recommend that stroke survivors are active every day, with physical activity volume gradually progressing from low intensity so that 150 min or more of moderate intensity physical activity is achieved per week [8]. Muscle strengthening activities are recommended at least twice per week, as well as activities that improve balance and co-ordination to reduce risk of falls. In light of the importance of exercise, and the challenges for health providers to facilitate opportunities for exercise with this population, it is important for researchers to examine targeted community-based, outpatient physical activity programmes for individuals living with stroke.

A number of qualitative studies have examined people’s experiences of living with stroke. This research has identified challenges faced when re-entering family life after a stroke [9], difficulties faced when engaging with health services [10], and problems encountered when returning to work [11,12]. For example, Pluta et al. [11] found working-age individuals living with stroke felt marginalised from their work life and this led to a sense of loss from their ‘previous life’ before the stroke. With regard to exploring experiences and long term needs when engaging with health services, findings illustrated how patients often felt they were not fully understood by healthcare services, with the authors suggesting healthcare professionals need to understand the individual needs of patients to allow them to successfully tailor support and set goals [10]. Researchers have also considered how to improve stoke care and highlighted how the needs of service users did not appear to be adequately addressed, with a lack of information and support given to patients [13]. These studies develop our understanding of some of the challenges faced by those living with stroke, thus providing an insight into how care might more effectively be offered in this recovery phase.

When directly considering physical activity, a small number of studies have looked at the exercise experiences of individuals living with stroke. Such research has included a focus on patients’ motivation to exercise and the facilitators and barriers to exercise. For example, in a study with individuals recovering from stroke, social interaction, beliefs of benefits of exercise, high self-efficacy, and the necessity of routine behaviours were found to be the most commonly reported motivators [14]. In addition, lack of professional support on discharge from hospital and follow-up, as well as transport issues to structured classes or interventions, and lack of control and negative affect were the most commonly reported barriers. Research exploring a community-based rehabilitation programme in rural Australia found stroke-specific exercise groups led by health professionals were viewed positively because they enabled social support, increased confidence, improved mood and motivation, and provided an opportunity to acquire knowledge from others [15]. Moreover, group settings have been found to increase exercise participation for individuals living with stroke due to the social support and encouragement people receive from other group members [16]. In research conducted with longer-term stroke survivors, findings have shown that promoting the psychological well-being benefits of exercise, such as increased self-esteem and life satisfaction, and offering different activity formats in de-medicalised settings, increase participation for long-term stroke survivors [17]. Such an increase in activity was explained by exercise being viewed as a healthy lifestyle activity rather than a treatment, with the latter potentially perceived more negatively. Understanding these motivating factors, facilitators, and barriers to exercise is important to consider for those healthcare professionals who aim to promote physical activity participation for those living with stroke.

In considering the literature around exercise provision for stroke survivors, Young and colleagues systematically reviewed the research on venue-based exercise programmes [18]. Findings of this review revealed that those living with stroke gain confidence and renewed identity through exercise participation, with participants reporting to enjoy stroke-specific exercise programmes in non-medical venues. Young et al. further suggested there is limited quality research in this area, highlighting a need to conduct research to examine stroke specific programmes, and furthermore, to consider ongoing, community-based exercise programmes designed specifically for individuals living with stroke. Our study therefore aims to build on the existing literature by considering the core research question of what are the experiences of people living with stroke who engage in a targeted exercise programme, in order to provide a more focussed analysis of the physical activity experiences of this clinical group. More specifically, we explore the experiences of individuals who engaged in a specific community physical activity programme, namely the HELP (Health Enhancing Lifestyle Programme) Hampshire Stroke Clinic. This programme was created to offer targeted physical activity and education interventions to individuals living with stroke following discharge from the NHS care pathway. In considering the experiences of these individuals, we aim to consider their perspectives of the delivery of this programme, and how it might support their rehabilitation and recovery from stroke. It is hoped that the findings from this study will allow us to more fully understand how exercise interventions for individuals living with stroke can be effectively developed, and in turn, provide healthcare providers with an underpinning evidence base to further develop and improve such exercise provision to support the recovery of those living with stroke.

## 2. Materials and Methods

### 2.1. Context, Design, and Philosophical Underpinning

This study focusses on exploring the perceptions of participants engaged in a local community-based exercise programme. The HELP Hampshire stroke clinic has a stated aim of enhancing the “physical and social quality of life for people living with stroke by reducing the likelihood of secondary stroke through low-cost, flexible, community-based exercise clinics”. We accessed participants who had joined the programme following either GP or hospital (via stroke consultants, physiotherapists, or occupational therapists) referral, with healthcare professionals (e.g., physiotherapists) and the community team conducting physical and psychosocial assessments at the start and following 12 weeks of exercise. There is one clinic which provided three weekly exercise sessions at the time the study was conducted. Participants had the opportunity to engage in weekly, group-based lower- and higher-intensity exercise classes, as well as Pilates. Each of these classes lasted one hour. Lower-intensity exercise classes involved seated and standing activities, including balance and co-ordination exercises using resistance bands and light weights, with participants following demonstrations provided by the lead instructor. Higher-intensity exercise classes included circuit-type activities, with weights and resistance bands, and periods of activity interspersed with short recovery periods and/or different types of active recovery. Each exercise class was group-based and cost £5 per participant. The lower-intensity exercise class tended to have 17 to 25 attendees per session, the higher-intensity class had between 12 and 15 attendees, while the Pilates class was capped at 10 people. Each exercise class had a lead practitioner, and was supported by volunteers at a ratio of 1 assistant practitioner for every 5 participants. The venues for the exercise classes were based in the community (not hospital or university site), in village halls, church halls and community centres. After completing the initial 12 weeks of the programme, participants were encouraged to either continue with the HELP Hampshire community-based exercise classes, and/or were provided guidance on other exercise and physical activity opportunities within their local community. A qualitative research design was used in the present study. Specifically, members of the research team conducted semi-structured interviews with participants to explore their experiences of engaging with the community exercise programme. A phenomenological methodology was used, with this approach aiming to “to arrive at a rigorous description of human life as it is lived and reflected on in its first-person concreteness” (p. 309) [19]. Such an approach is concerned with understanding experiences from the perspective of the individual and, in the current study, a phenomenological approach was appropriate to explore the experiences of individual participants within a complex social world while they were living with stroke. In line with our aims, the research approach was underpinned by ontological relativism and epistemological constructionism as we encouraged participants to share their individually constructed reality of living with stroke and participating in an exercise programme designed to support their rehabilitation. We also acknowledge the active role the authors played in the co-construction of knowledge of a phenomenon that cannot be directly observed, through the collection, analysis, and presentation of data.

### 2.2. Participants, Procedure and Interview Guide

Following favourable university ethical review at the lead author’s university, and based on purposive sampling procedures, participants from the community exercise programme were invited to participate in the project. The inclusion criteria for the study were that participants were living with stroke and had attended, on average, at least one HELP Hampshire exercise class each week for a period of 12 weeks. Thirty-five participants attending the exercise sessions who met the criteria were informed of the study, and invited to participate in an interview, with 16 agreeing to participate. These invitations were extended either by explaining the study in person at the end of classes or by email, with initial email contact being made by one of the project leads. If individuals expressed an interest in participating, they were contacted by phone or email to discuss details and arrange an interview. The interviews took place either at a time and location suitable to the participant or at an agreed time online. All interviews were conducted either by the second and third authors who were independent of the delivery of the exercise classes. A semi-structured interview approach was adopted, with an interview guide developed that included open questions, to encourage participants to talk freely about their personal experiences of stroke and the local community programme and, in particular, about the group-based exercise classes. Initial questions were asked to develop rapport and included questions about the participant’s wellbeing and family life since their stroke, before moving on to asking questions about their general activity levels (e.g., “tell me about your current physical activity” and “how has your activity changed since the stroke?”). 

Moving into the main part of the interview, we considered other research that has previously explored participant experiences of exercise interventions (e.g., [14,17]) to develop further questions. This part of the interview started with a broad open question of “Can you describe your recent experiences of physical activity as part of the HELP Hampshire clinic sessions”, which allowed participants to take control of the interview and guide the conversation in whichever way they wanted about the programme. Further specific prompt questions were asked (e.g., about the types of exercises or the people they met), and probing questions were also utilised to encourage rich, detailed explanations (e.g., “how did that make you feel?” or “can you say a bit more about that?”). Some participants had speech difficulties and, thus, the researcher used paraphrasing to check and confirm understanding of what had been said. Furthermore, when piloting the interview process, it became apparent participants found difficultly in answering some questions, such as specific questions asking them to recall specific exercises they had done in the class. As a result, amendments were made to the phrasing of questions to focus more on reflecting on their general experiences, feelings, and emotions around being involved in the classes (which participants found much easier to give detailed responses to). Where appropriate, a pen and paper were also provided for participants to write something down. The interviews continued until a natural close as the interviewer tried to judge fatigue levels of the participants and to take into account that a small number of the participants had memory and communication difficulties which made it difficult to talk at length. 

### 2.3. Data Processing, Analysis, and Rigour

Interviews lasted between 15 and 67 min (average 46 min) and were recorded and transcribed verbatim. Using a reflexive thematic analysis, the researchers moved back and forth through several stages of analysis, including data familiarisation and coding, theme development, refinement and naming and writing up [20]. This process initially involved the first author immersing themselves in the data by reading and re-reading the interview transcripts, before deriving initial open codes, highlighting features of the experiences of the exercise classes described by participants. The process of analysis was data-driven, with transcripts inductively analysed and coding based on the data, rather than deductively coding data based on existing frameworks [21]. Once all of the transcripts had been thoroughly coded, the codes were sorted into potential themes that represented “some level of patterned response or meaning” across the data set (p. 82) [22], relevant to the research question. Theme development involved clustering the codes using visual mind-maps to illustrate how codes might link together into overall themes. As a result of these analyses, the codes were clarified into overarching themes and sub-themes, with four overarching themes developed from the data. To enhance the analysis process, the second and fourth authors acted as critical friends, challenging decisions made on the development of themes [23]. For example, one critical friend highlighted the importance of considering the different sub-themes within the ‘personal progress’ theme to represent the progress that participants both see directly and also feel within themselves. This reflective process involved discussion and debate to encourage greater exploration of alternative interpretations of the data analysis [23].

## 3. Results

Table 1 illustrates the demographic characteristics of the 35 participants invited to interview. Of the 16 participants recruited, eight were living with weakness on the right-side, and eight with weakness on the left-side. Within the sample, 12 participants had experienced a first-time stroke/TIA, and four participants had experienced two or more strokes/TIAs. On average, participants had been involved in the exercise programme for 15.6 ± 3.6 weeks at the time of invitation to the interview, and on average, the interviews took place 17 ± 15 months since the stroke had occurred. 

From our analysis, four overarching themes were developed: (i) Feelings of Appreciation, (ii) Interactions with Other Patients, (iii) Positive Contributions of Trained Instructors, and (iv) Personal Progress. These results are summarised in Table 2. In the following section, the four overarching themes are presented with accompanying illustrative quotes to represent each theme. Within these four overarching themes, there are 13 sub-themes, with a sub-theme included in each overarching theme outlining the suggestions that participants made that might improve and develop the programme.

### 3.1. Feelings of Appreciation

This first overarching theme represents how the participants expressed appreciation for the programme, being grateful of how it offered them the opportunity to exercise. In addition, the participants gratitude appeared to come as a result of a feeling that there was a lack of care before this programme offered them the chance to join the exercise programme.

#### 3.1.1. General Gratitude (for the Opportunity to Exercise)

Participants made a variety of positive comments that highlighted their gratitude for the programme. Participant 2 said “I think what [the project leader] is doing is excellent” and participant 4 commented “I think we’re very fortunate...to be on this programme”. Participant 2 highlighted some of the benefits of having the opportunity to exercise saying “I think the fact that we’ve got these classes to go to is good for us because it does, you know, keeps your heart going, and the fact that occasionally you have an assessment, I think that’s good”. Participant 8 summed up their gratitude for what the programme offers by explaining.

It’s given me something to look forward to, I mean, I’ve got a few things to look forward to in my life, thankfully, but, you know, on a weekly basis it’s good to have something that forces you to get up on a Monday morning, you know, and, you know, something to go to, structured. I think it’s good.

#### 3.1.2. Appreciation (Feeling That Nobody Cared before This)

Participants appeared particularly grateful for the programme, as it gave them the feeling that somebody cared about them. Participant 2 said “All credit to them for doing this, because before [the project lead] did this, after you’ve had a stroke, nobody cares, do they”. Participant 8 explained further that after initial support from healthcare providers following their stroke, they felt a certain feeling of being deserted; “you’re on your own. Yes, you’ve got your GP, but basically, that’s it. You are…here are your pills, keep taking these, and we don’t want to see you again. And it’s like, well…”. This participant highlighted how “I’m still not a well person” and said about the programme that “… it means a lot. Means a great deal. You know, you don’t feel as if you’re just left to, you know, abandoned, really”.

#### 3.1.3. To Improve the Programme

Participants made some general suggestions around how the programme could be improved. For example, participant 2 would have liked the communication between the those delivering the programme and the patients to be improved, suggesting wanting “them to email us occasionally if things are going to be changed”. Participant 8 commented on enjoying the classes, but also mentioned how they would like a greater variety of activities in the classes, such as playing games; “I love ball games…the thing about ball games is that, you know, you do various kinds of movements, it’s coordination, it’s balance, it’s strength, it’s movement, it’s aerobic, you know, it’s everything. And it’s fun!”.

### 3.2. Interactions with Other Patients

The second overarching theme is around the opportunity for interactions with others living with stroke. This includes the general social interactions that the programme promotes as well as specific positive feelings resulting from interacting with other patients who are similar to them.

#### 3.2.1. Developing Positive (Social) Interactions

Many comments were made by participants about how they enjoyed the interactions with others that attending the exercise classes allowed them to have. Participant 13 highlighted how all the “small interactions” they had “are quite meaningful”. Participant 9 described the “togetherness” created by being in “a little community”. Participant 3 gave an example of a basic interaction at the start of the class, recalling “when I meet him I say ‘how are you?’ or whatever his name is…’Bloody awful’ she says, laughing. And I think that’s good for her”. Participant 13 also spoke about the types of general interactions that occurred, “you tended to interact and meet with different people so it was quite easy to then say ‘oh hello I remember you from last week oh and how’s this’ blah blah blah you know”. Participant 6 highlighted how these interactions had a positive impact, saying “My mood is much better now. We have a laugh about things. You know me with [another patient], we will have a laugh there”.

One participant (5), who had some speech difficulties, and struggled to articulate their thoughts fully, spoke about the friends they had met, “But, friends, friends, friends, yeah, yeah”). When the interviewer prompted him saying “And what is it you like most about coming to the class?”, this participant responded enthusiastically by saying “friends, yeah”. Interestingly, some participants spoke about creating social groups with those in the exercise classes, where they would meet outside the class time, for example, meeting at cafes, or participant 5 also talked about how five members of the group would meet at one of their houses to socialise.

One point to highlight is that the importance of such interactions depends on the personality of the participant. Participant 12, explained “I don’t think that I would go to it as a social gathering of any kind, because I’m not really that kind of person so to me it’s not something that’s appealing to me”. 

#### 3.2.2. Being with Others Who Understand

Further to mentioning the general positive interactions that the classes created, participants also commented on the positivity of these interactions being with others who shared similar experiences. Thus, these interactions had a greater resonance as they were with people who had a deeper understanding of what they were going through. Participant 3 spoke about “meeting other people in a similar situation and sharing thoughts”, and participant 4 talked about;

Being with others who know what you’re going through, and being able to just have that conversation with other people and talk about things and…be on the same level. If you see what I mean? Knowing about what you’re talking about…that was a big benefit to the programme, actually. 

Such experiences were supported by participant 13, who illustrated the importance of being in the class with others who were living with stroke;

It was specifically to do with having a stroke. It’s not like you know you have those same, relatively similar conversations in an ordinary class that you go to, but they could be about the weather or, you know, would be brief, conversations whereas with this it was you knew that the unifying factor was everybody at some point had had a stroke. And that was why it was probably made it easier to sort of to mention, you know, to sort of bring things up.

Participant 8 also commented on how it was “good to see fellow sufferers, you know, to see how they’re getting on and encourage them”. This participant went on to explain;

We’re all similar. And, it’s just good to be able to see that you’re not on your own. And, there are other people who are experiencing the same thing, and I think we kind of encourage each other. I’d like to think we do. And, it’s good to compare notes, so to speak, on all sorts of things, really, like medication, for example.

#### 3.2.3. Suggestions to Improve Interactions

Certain specific suggestions were made by participants that they felt would enhance the potential for interactions with others. For example, participant 7 proposed how developing the social aspect would improve the experience further, “they could make it a bit more of a social thing, it was useful just talking to people who are in the same situation and realising, okay, actually we’re all going through the same thing”. This was reinforced by participant 4 who explained how structured opportunities for interactions would be good;

Interactions were important, classes could have been improved by giving opportunity to talk with like-minded people, with time factored into those classes to perhaps just sit down and have a have a chat at the end of the session…it would have been really interesting to listen to what happened to other people…And how perhaps they’d overcome different difficulties, etc.

A further participant (2), who said they found it “difficult to remember names” commented, “I liked when we threw balls at each other…but it’s remembering people’s names, isn’t it…it would be good to have a name on people, you feel so stupid when you forget”. A further suggestion was made by participant 8 who proposed the idea of a WhatsApp group to encourage further interaction and support, saying, “I’ve been a bit surprised that there isn’t an email group, or a WhatsApp group, to, kind of, build a greater sense of community, if you like, to sort of encourage each other”.

### 3.3. Positive Contributions of Trained Instructors

The third overarching theme refers to participants reflecting on the contributions of the stroke-specific-trained exercise professionals who facilitated the exercise programme. Participant 16 commented that “without that support I would fail miserably. And so, for them to be around. It’s been a great help”. Participants also made numerous positive comments about the personal qualities of the instructors and staff who delivered the programme. For example, comments included how instructors were “constantly enthusiastic” (participant 13), how the staff were “always smiling” and they “feel very happy to see them” (participant 6) and how instructors are “very good at motivating” (participant 12). This theme included two further sub-themes, which included the quality of instruction offered, in terms of the way instructors offered support and reassurance around exercising, and a progressive coaching approach that supported them in their rehabilitation.

#### 3.3.1. Reassurance about Exercise Adaptations

Participants spoke enthusiastically about the support the instructors offered. For example, participant 7 highlighted the reassurance they gained from such support, “Just knowing that I was in an environment where it was controlled and with people who actually know what they’re talking about”. Participant 11 expressed the trust they had in the instructors and the advice they gave, saying, “It is, valuable it is valuable. I can’t explain, I don’t know how to explain it. But it is valuable. When they say jump, I ask how high”. Participant 15 reinforced this by providing specific examples of such advice and support;

[Instructor’s name] is always saying, have a support near you, and for this exercise, “keep the support like a chair on your righthand side”. She says if you’re feeling a bit tired, don’t do it standing up, do it sitting down, and she demonstrates the whole thing.

Participant 4 commented “It’s useful to have people there assessing us all the time, watching what we’re doing, because we could damage ourselves especially if we were doing something [we’re not sure about]”. This participant also reflected on the positive impact such support had on their approach to exercise;

I was thinking, ‘Alright, I can trust you if you say I can do all these things’. And, even in the first week it made a big difference, because I was really nervous about just looking over my shoulder because of the damage to my artery.

Participant 13 appreciated the support to overcome discomfort associate with the exercises. It also appears clear from this quote the confidence the participant felt from having such specialist advice about exercises, and in this case, how it reassured them that the pain experienced was a normal part of the rehabilitation process. 

[Instructor’s name} just gives you exercises to try and give you the confidence to know it’s okay if you do this, that you’re not going to do yourself any harm, in fact that the opposite is the case that you’re going to make it, it’s going to be easier if you sort of almost go through the pain barrier to start. We were doing an exercise and I just wobbled about like a complete muppet but it’s, it’s fine, because it gets better.

#### 3.3.2. Progressive Coaching Approach

Further to the advice and support that the instructors provided, participants commented on the direction provided to help them to progress with their rehabilitation. Participant 13 commented on the specific advice given to help them recovery effectively;

They said if you do walking forward sideways, then stretch, then you’ll soon be able to do, you know, in spite of having a stroke you’ll be able to do more things. It sort of just tips it on its head so that you’re not being passive in your recovery. 

Participant 15 provided an example of such an approach that supported progress and recovery.

[Instructor’s name] will say “do as many as you can in this one minute”. And I can count to, maybe 35, then they’ll say ‘okay, this session concentrate on posture and slow down. Steady back up again putting more of a correct movement into the exercise’. And that number goes down to maybe 27, 28, and I can work up from that. So there is immediate feedback because I know how many I can do in a minute under what criteria be it fast or doing the posture correctly.

#### 3.3.3. To Improve (Contributions from Instructors)

Participants offered suggestions about the role of the instructors and how sessions might be developed further. Participant 2, who acknowledged they had been very sporty before their stroke, would ideally like greater variety in the sessions, commenting, “I’ve always played golf and tennis and hockey and everything…I know it’s not everybody’s sort of thing. But doing that same old thing in the circuit, I find just a bit boring”. This participant also suggested the possibility of playing certain games to allow them to “to use our minds a bit, you know, use their memory, you know, because I think if your memory is going, maybe to stimulate it more”. Participant 9 suggested how the programme might link with other professionals (e.g., nutritionists) to provide individualised advice, highlighting how “you almost need a sort of a partner, who’s following you around and from a medical perspective and from a whole lifestyle perspective saying, okay, well, we can work out this for you”. A final suggestion was from participant 3 who asked if there might be opportunities for more individual time with the instructors, saying “…just to give some time. I know they say, frequently, “now if anyone’s got any problems, talk to us”, but they’re all clearing up, and they’re all wanting to get back to their next job”.

### 3.4. Personal Progress

The final overarching theme is around the progress participants felt they were making as a result of engaging with the exercise programme. This theme is broken down into sub-themes that include both seeing and feeling they are making progress, as well as identifying the importance of progressing with others.

#### 3.4.1. Seeing Progress

Numerous participants gave examples of seeing definite progress in their recovery. Participants 6 talked about how they were feeling “more stable”, participant 13 said about how they had “noticed I still have issues surrounding balance. That’s improved a lot through, you know obviously having the classes”, and participant 4 reflected how the programme had made them feel “physically stronger…which has boosted my morale…it’s made me realise that I can improve. That’s obviously good. And I am improving week on week”. Participants provided further specific examples of the improvements that they saw in themselves. Participant 1 commented how they “can do squats a lot better… all the exercises made me stronger and a little bit more mobile”. Participants spoke about how the progress they had made was evident in tasks they were now able to complete, with participant 15 enthusing about this: “Yes, just everyday stuff. Cutting the lawn, when it first came out, I couldn’t do the whole thing in one go, but now I could do front and back garden in one go. Just looking back at general progress across the board”. Supporting this, participant 7 highlighted how the exercise programme “really has made a big difference. Yeah. Because, before, like I said, I could go to London for a day and then be wiped out. I never would have considered doing the stuff I’m doing now”.

#### 3.4.2. Feeling Progress

Participants also spoke about how they could feel the progress they were making. For example, participant 16 spoke about how the exercise classes “make you feel a lot better. Yeah, in so much as that it gives you satisfaction that you’re able to increase the number of exercises”. Participant 16 went on to further explain how “mentally, I feel a lot better, because of the exercises that carried out and also physically that you do feel better, because the. It loosens everything up for you”. Participant 8 spoke about some of the tasks they were now able to complete in their everyday life and commented on this positive impact, “not only [reduced] fatigue, but in my confidence and knowing I could do things and not have to worry about and thinking ‘oh, should I’”.

#### 3.4.3. Progressing Together

When speaking about progress, a number of participants mentioned the importance of not being on their own, and the value of being with similar others and seeing them progress also. Participant 3 explained; 

Oh, it’s a blessing. Not only, was it good for me because I saw other people improving, you know? Some of them, you know, were in a bit of a state but they’re gradually improving…some seem to struggle, but you can see them improving…and that helps me. 

#### 3.4.4. To Improve (Personal Progress) 

The only real suggestion participants made about improving progress, was participant 12 who mentioned that after exercise classes, something could be recorded (e.g., written down) about the specific exercises. This participant highlighted how this would be beneficial;

That would help you to remember to what’s next, you’d need a timer to time yourself with the exercise and you need to be told what you need for the exercise like whether you’re going to use weights or whether you’re going to use a band or something like that.

## 4. Discussion

The aim of this study was to explore the experiences of people living with stroke who had engaged in a targeted community physical activity programme. The principal finding was that the participants spoke very positively about the programme, expressing gratitude and appreciation for the availability of the classes and the opportunity to exercise while supervised by trained professional staff. Findings also highlighted how participants valued the role of the professional facilitators, who gave support, encouragement, and specialised instruction; how participants recognised the importance of the social interactions created from participating in the exercise classes, particularly with others living with stroke who shared their experiences; and the merit of seeing the physical progress they were making.

Previous research that has examined exercise provision for those living with stroke, has highlighted how the needs of service users have been seen to not be adequately addressed, with a lack of information and support given to patients [13]. Our results were in contrast to these findings, with participants in our study highlighting the high-quality support offered by the class facilitators, which included a high level of instruction. Indeed, previous research exploring therapists’ experiences of developing physical activity opportunities for residents in care homes found developing trusting relationships between staff and residents was key in intervention work to promote physical activity [24]. In line with these findings, trusting relationships between facilitators and participants in our study appeared to be evident in the responses of the participants. In addition, and reflecting previous research (e.g., [14,25]), the results of our study indicate that professionally trained instructors and the interactions experienced are perceived positively by our participants. These elements appear to develop feelings of confidence in the participants, increasing their self-efficacy towards involvement in the classes. Self-efficacy is the main construct of Bandura’s Social Cognitive Theory [26], which is underpinned by interactions between personal, behavioural and environmental influences. Two of the main steps for building self-efficacy are argued to be mastery experience and vicarious experience, which align with our findings. Participants highlighted the specific instruction that facilitators gave to help them succeed in the exercise tasks, which created mastery experiences in exercising. Seeing progress in their physical recovery as a result of the classes is also likely to contribute to such feelings of mastery. In addition, vicarious experience can have a positive impact on a person’s self-efficacy for physical activity [27,28]. Our participants also explained how being in a group with similar others, and seeing their progress was perceived very positively. Taken together, it appears the efficacy beliefs of participants are enhanced by features of the exercise programme.

Our findings illustrate the importance of the interactions that were created by attending the exercise sessions, with the majority of participants speaking about the importance the of developing such relationships. In a meta-synthesis of qualitative literature that has examined stroke rehabilitation and community services, results include a ‘social environment’ theme that notes how group exercise and being with other stroke survivors can enhance positive interactions and provide the social support that is important to stroke survivors [29]. Our data highlighting the importance of social interactions support the findings of this meta-analysis, and other research that illustrates the value of developing interactions with others [14,15]. For example, Nicholson et al. found that social interaction was a key motivator for stroke patients being physically active [14], and our findings indicate that such interactions were perceived very positively by our participants. A further explanation for the importance of interactions might be from the support that stroke patients offer each other as a result of the relationships developed. This aligns with previous research, which found that group settings increase levels of exercise participation for stroke patients, due to the social support provided and encouragement received from other group members [16]. To improve the experience further, our participants also suggested that further time in exercise sessions should be allocated for directly engaging and speaking to each other. It seems our findings suggest that a key positive consequence of exercise programmes is to create and promote connections between participants within the exercise class settings.

Not only did participants highlight the importance of relationships developed, but furthermore, they emphasised the importance of engaging with others who also lived with stroke, and thus, had a shared understanding of each other’s situations. This appears to suggest that the participants developed a strong sense of identity from being part of the classes with people they could relate to, which was valuable to them. Social identities are proposed to occur in situations when individuals can identify with others in their group, and in turn, group membership becomes important to who they are as people [30]. The multidimensional construct of social identity involves the significance of being a group member, the positive emotions resulting from this group membership, and the strength of connections within the group [31]. Our findings highlight the positive responses of the participants that illustrate the importance of being part of the exercise classes with similar others and thus, the importance they attach to membership of the group. Indeed, identity theory posits that individuals derive a sense of who they are from their group membership [32], and these positive outcomes as a result of membership of the stroke exercise clinic were very evident in participant responses. Suggestions from participants also indicated that those running exercise programmes might develop additional ways to promote and strengthen feelings of identity and group membership to encourage further interactions (e.g., a WhatsApp group and further intra-group support). 

### Limitations and Suggestions for Future Research

Various limitations of our research should be acknowledged. Firstly, we might consider the individual differences of those who were interviewed. Within our sample there were participants who were already quite active before their stroke. Thus, some participants might be more likely to engage positively with exercise programmes anyway, and might be more likely to be positive about targeted programmes to allow them to exercise further. Future research might explore the experiences in more depth of a specific sample who engaged in limited exercise before their stroke. To also develop this research, presenting our findings back to either our participants, or a further sample of stroke patients would act as a member reflection process [23] to further explore how the findings might resonate with them. In addition, our participants may have been more extrovert and open to experience [33], and thus, be more likely to enthuse about opportunities for interactions within the classes. However, we did not capture the reasons why certain invited participants (*n* = 19) chose not to participate in the interviews. Future research might use cross-sectional designs that consider personality characteristics when examining predictions of motivation and adherence towards community exercise classes. 

We also recognise that some participants had speech difficulties that impacted on the interview process in places. For example, a participant commented on their trust in the instructions of the facilitators, and when prompted to consider why they felt this way, they struggled to articulate why this was. Future research might use ethnography and observational methods to supplement data collection with those with speech difficulties and thus, help understand participant experiences further. Future research might also consider the use of methods such as ‘Talking Mats’ [34], whereby people with speech and communication difficulties can be enabled to discuss quality of life using picture communication symbols linked to the topic guide and visually recorded instead of just audio-recorded. Future research could also consider using more creative methods such as photo elicitation [35] or videos from the exercise classes to stimulate memories and discussion around their experiences. In addition, much literature is based on single interviews, that explore exercise referral programmes that have short-term duration, as in the case of the current study, with long-term involvement under-investigated [5]. Future research might interview those living with stroke at multiple timepoints over a longer period to provide a more thorough view of their exercise experiences. Finally, while the suggestions made in this paper have not been directly used to refine the programme thus far, future research might examine how the exercise classes could be developed and enhanced in light of these suggestions.

## 5. Practical Implications and Conclusions

The findings of the current study suggest that participants have very positive perceptions of the HELP Hampshire exercise programme, and are very grateful for the opportunity that the exercise classes provide. In terms of practical suggestions for healthcare providers who might develop similar programmes for clinical populations, our results lead to three specific recommendations. First, participants identified the role of the trained exercise professionals who facilitated the sessions, who provided support, encouragement and specialised instruction. Thus, programme project leads need to recruit expert facilitators who can effectively meet the complex and heterogenous needs of individuals living with stroke. Second, participants highlighted the importance of the social relationships that result from belonging to exercise groups, and exercise programmes should look to foster and encourage interactions with others living with stroke, which can enhance feelings of belonging and a sense of identity within the group. Third, participants were enthused by seeing progress in their recovery, and exercise programmes should look to include a range of formal (e.g., structured assessments) and informal (e.g., regular feedback from facilitators) methods to enhance awareness of the progress they are making while participating in the classes.

Previous research has explored the experiences and long-term needs of those living with stroke [10] and highlighted the need for healthcare professionals to understand the specific needs of patients in order to effectively tailor support and set goals. Furthermore, Martinsen et al. suggested that support programmes should be set in collaboration with patients. The present study has explored the perceptions of stroke patients on a community-based exercise programme, and we hope that our findings help to further our understanding of the needs and experiences of those living with stroke who engage with such exercise programmes. In turn, we hope the findings can be used to provide an evidence base to underpin the development of other exercise programmes both for those living with stroke or other long-term conditions.

## Figures and Tables

**Table 1 healthcare-11-00154-t001:** Participant and non-participant characteristics.

		Participants *n* (%) or Mean (SD)	Non-Participants *n* (%) or Mean (SD)
Number		16	19
Sex	Male	10 (63%)	13 (68%)
	Female	6 (37%)	6 (32%)
Age (year)		64.6 ± 10.2	62.5 ± 8.6
Age range (year)		51–85	45–80
Body Mass Index (kg·m^2^)		26.9 ± 5.0	27.7 ± 6.2
First time stroke/TIA		12 (75%)	13 (68%)
Ischaemic Stroke		14 (88%)	16 (84%)
Time since first stroke (months)		17 ± 15	14 ± 17
Symptomatic side	Right	8 (50%)	11 (58%)
	Left	8 (50%)	8 (42%)
Use of a walking aid		6 (37%)	5 (26%)
Use of a wheelchair		0 (0%)	1 (5%)
Average time participating in programme at time of invitation to interview		15.6 ± 3.6 weeks	14.5 ± 4.2 weeks

**Table 2 healthcare-11-00154-t002:** Overarching themes, subthemes and sample raw data themes.

Overarching Themes	Sub-Themes	Sample Raw Data Themes
Feelings of appreciation	General gratitude (for the opportunity to exercise)	- It’s good for us - Something to look forwards to.
	Appreciation (feeling that nobody cared before this)	- After a stroke, nobody else cares - You feel abandoned otherwise
	To improve (the programme in general)	- Improved communication - More variety in classes
Interactions with other patients	Developing positive (social) interactions	- Togetherness in a small community - Friendships developed
	Being with others who understand	- Others know what you’re going through. - Not feeling on your own - Stroke the unifying factor
	To improve (interactions)	- Further opportunities to interact in class. - Social media group to encourage interaction outside of class.
Positive contributions of trained instructors	Reassurance about exercise adaptations	- Trust in instructors - Showing correct technique - Offsetting nerves about exercising with specific injuries.
	Progressive coaching approach	- Helping achieve greater progress. - Focussing on correct movements.
	To improve (contributions from instructors)	- Greater variety in sessions. - Links with other professionals. - Time to talk with instructors.
Personal progress	Seeing progress	- Greater balance - Physically stronger - Accomplish tasks outside class.
	Feeling progress	- Satisfaction with progress. - Confidence in recovery
	Progressing together	- Others improving provides motivation.
	Suggestions to provide further support	- Instructions written down.

## Data Availability

The data presented in this study are available on request from the corresponding author. The data are not publicly available due to participants’ privacy protection.

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
