# Peer review of "Understanding the Experiences of People Living with Stroke Engaging in a Community-Based Physical-Activity Programme"

_healthcare, 2023, doi:10.3390/healthcare11020154_

Round 1
Reviewer 1 Report
The article raises an interesting topic that could provide rich information towards progress in the quality of life of people with stroke. However, I recommend one series of improvements that could offer greater scientific rigor to the work.
1. First, who’s experiences in the title? Not clear.
2. It’s very important to conduct a qualitative research based a theoretical framework. It would be convenient to introduce contributions from current scientific works on the importance of sport or physical activities in improving the quality of life of people with stroke by using the theoretical framework. The authors used some views of philosophies, such as interpretivism, ontological relativism, and epistemological constructionism, to understand participants’ experiences. I could not get this meaning. I think these philosophies could not guide this study.
3. In the methodology section more information is required on the process followed for the construction of the interviews, the categorization process and the triangulation of the results obtained.
4. Possibly, getting more detail about the methodological process can improve the results and reach more reliable conclusions
Author Response
Please see attached file with full response to reviewer's comments

Reviewer 2 Report
Understanding the experiences of engaging in a community- based, physical-activity focused secondary stroke prevention program
Matthew Smith*, Andrew Scott, Serena Mellish and James Faulkner
Title: The title includes the terms “physical-activity focused secondary stroke prevention program”. The reader needs to know how the program focused on preventing secondary stroke if that phrase is in the title. However, occurrence of secondary stroke is not evaluated. Consider editing the title.
Purpose from Abstract: “Our study aims to explore the physical activity experience of people living with stroke within a community physical activity programme.” I believe you mean your aim was to explore the feelings people living with stroke had about the HELP (Health Enhancing Lifestyle Programme). The structured interview does not bring out comments about the physical activitiy – how hard it was, was it enjoyable, did it hurt, did they experience any injuries? The semi-structured interview did record social experiences.
“This programme was created to offer targeted physical activity and education interventions following the discharge of patients from the healthcare pathway.” A list of the targeted physical activities and education interventions provided at HELP would be helpful to the reader. Did the program provide suggestions for continued physical activity and education post HELP. Was the HELP (Health Enhancing Lifestyle Programme) Hampshire Stroke Clinic program free?
This qualitative study involved semi-structured interviews with participants living with stroke who were recruited from individuals who had engaged with the activity programme. Please tell us in the Methods section more about the selections of participants. How many were asked to participate? How many agreed? How did you decide whom to ask? Pretesting and post testing: was conducted but not reported. “team conducting physical and psycho-social assessments at the start and on completion of a 12-week exercise programme”. The reader should know participants conditions pre- program since there is no Control group. Please justify the Research Design used. “A qualitative research design was used” The reader needs to know more.
Summary of patient population: Sixteen participants is a low number of participants. 4 F, 12 M. Can you justify how 16 participants was adequate? “Based on purposive sampling principles” – this phrase is not enough.
In the Methods Section, the reader wants a summary of post-stroke participant characteristics: pathology, age range, body mass, number of strokes each, use of aids (hearing, eye) walkers, braces, etc; mobility. Average number of classes patients participated in per week. Types of classes. Was all instruction by group class? How was transportation to class achieved?
How many patients were invited to participate? Were all interviews in person? What were the lengths between ending HELP and conducting the interviews (Mean + SD).
The HELP clinic (are there more than 1 clinic) For this study, was the program organized specifically for these participants or were these participants selected from all patients who attended classes at HELP? What were the number of patients per session? All post stroke patients? Mixed group? Instructors/patients ratio. What makes HELP a “venue based program”?
Please justify the use of this one group research design. It is fine to acquire qualitative – descriptive data if the design is justified. Using phenomenological methodology is fine when the design is accepted and the study group is well defined and justified.
“phenomenological methodology was used, with this approach aiming to “to arrive at a rigorous description of human life as it is lived and reflected on in its first-person concreteness”. This sentence is very general. Can you be more specific?
Scheme and Programme seem to be used interchangeably. Help the reader know the difference.
The HELP clinic has a community exercise scheme. The manuscript uses the term scheme often “Patients join the scheme”. It becomes confusing because the reader is not clear if the scheme is a program before the data analysis or develops through the analysis or if data are fit to the scheme. Please clarify this early in the Introduction. “Four overarching themes” are then reported within the Scheme. The Scheme referrals reads as Circular – what came first – what developed after listening to interviews?
Was the semi-structured interview appropriate for a group of post stroke patients before the physical activity and education interventions?
Was the Semi Structured Interview format published and validated with a specific population? Please tell the reader more about the number of number of questions.
“Findings of the review revealed that those living with stroke gain confidence and renewed identity through exercise participation, with participants reporting to enjoy stroke-specific exercise programmes in venues that were de-medicalised.” …. “to examine stroke specific programmes, and furthermore, that ongoing, community-based exercise programmes designed specifically for individuals living with stroke are required.”
Young et al, 2021.
This sentence is the purpose of your study. It needs editing. “Our study therefore aims to build on the existing literature by considering the core research question of what is the experience of people living with stroke of physical activity, in order to provide a more focussed analysis of the physical activity experiences of this clinical group.”
Again, this sentence needs editing. “More specifically, we aim to advance the literature by exploring the experiences of individuals living with stroke in a targeted community physical activity programme, the HELP (Health Enhancing Lifestyle Programme) Hampshire Stroke Clinic.”
In considering the experiences of these individuals, we aim to consider their perspectives of the delivery of this programme, and how it might support their rehabilitation and recovery from stroke. It is hoped that the findings from this study will allow us to more fully understand how exercise interventions for individuals living with stroke can be effectively developed, and in turn, provide health-care providers with an underpinning evidence base to further develop and improve such exercise provision to support the recovery of those living with stroke.
This sentence comes out of nowhere. Sleep was not evaluated in this study. Please remove this sentence. “Findings provide support for the first hypothesis that stressors would negatively predict quality of sleep”
Some typos.
Author Response
See attached document for full response to the reviewer's feedback

Round 2
Reviewer 1 Report
I think the authors did not reply my comments well.
The third comments : It’s very important to conduct a qualitative research based a theoretical framework. It would be convenient to introduce contributions from current scientific works on the importance of sport or physical activities in improving the quality of life of people with stroke by using the theoretical framework.
I think the authors did not understand this comments and refuse to revise. If like they said, they "deductively analyzing the data", the authors did not present the methodology in the method part. It should be ground theory. The authors refused to the think about the fourth comments.Therefore, I think this manuscript did not meet the publication demands in Healthcare.
Author Response
Dear Reviewer 1,
Thank you for engaging with our revisions to the manuscript. However, despite the extensive work undertaken to revise the manuscript, we were disappointed to read that you still were not happy with our amendments.
In the authors opinion, your comments are very subjective, and indeed lack a clarity and understanding of the qualitative approaches we have taken. For example, your comment “The authors used some views of philosophies, such as interpretivism, ontological relativism, and epistemological constructionism, to understand participants’ experiences. I could not get this meaning. I think these philosophies could not guide this study”, this indicates a lack of appreciation of the philosophical underpinning of qualitative research that is commonly outlined in qualitative manuscripts.
Your point about including a theoretical framework is also very subjective. This is not a ‘definite rule’ in qualitative research, and while it may be appropriate in some qualitative studies, we argue it is not needed in our study (as we are not testing theory).
Your comment about us ‘deductively analysing the data’ is simply not true. In the manuscript, we write “The process of analysis was data driven, with transcripts inductively analysed and coding based on the data, rather than deductively coding data based on existing frameworks”.
Your comment “It should be grounded theory” does not make sense to us. Grounded theory is a methodology designed to develop theory, which is a long way away from what we are aiming to do with this paper.
Reviewer 2 Report
The authors have significantly improved the the readability of their manuscript. The logic and methodology is now clear. There program for stroke patients deserves promulgation. I hope many read and learn from this work. Thank you for addressing this reviewer’s suggestions.
Author Response
Dear Reviewer 2,
We thank you for providing constructive and helpful feedback that allowed us to make detailed changes, and thus, greatly improving the manuscript.